# The Role of TRPM2 in Endothelial Function and Dysfunction

**DOI:** 10.3390/ijms22147635

**Published:** 2021-07-16

**Authors:** Wioletta Zielińska, Jan Zabrzyński, Maciej Gagat, Alina Grzanka

**Affiliations:** 1Department of Histology and Embryology, Faculty of Medicine, Collegium Medicum in Bydgoszcz, Nicolaus Copernicus University in Toruń, 85-092 Bydgoszcz, Poland; w.zielinska@cm.umk.pl (W.Z.); agrzanka@cm.umk.pl (A.G.); 2Department of General Orthopaedics, Musculoskeletal Oncology and Trauma Surgery, University of Medical Sciences, 61-701 Poznan, Poland; zabrzynski@gmail.com

**Keywords:** TRPM2, endothelial dysfunction, reactive oxygen species, calcium ions

## Abstract

The transient receptor potential (TRP) melastatin-like subfamily member 2 (TRPM2) is a non-selective calcium-permeable cation channel. It is expressed by many mammalian tissues, including bone marrow, spleen, lungs, heart, liver, neutrophils, and endothelial cells. The best-known mechanism of TRPM2 activation is related to the binding of ADP-ribose to the nudix-box sequence motif (NUDT9-H) in the C-terminal domain of the channel. In cells, the production of ADP-ribose is a result of increased oxidative stress. In the context of endothelial function, TRPM2-dependent calcium influx seems to be particularly interesting as it participates in the regulation of barrier function, cell death, cell migration, and angiogenesis. Any impairments of these functions may result in endothelial dysfunction observed in such conditions as atherosclerosis or hypertension. Thus, TRPM2 seems to be an attractive therapeutic target for the conditions connected with the increased production of reactive oxygen species. However, before the application of TRPM2 inhibitors will be possible, some issues need to be resolved. The main issues are the lack of specificity, poor membrane permeabilization, and low stability in in vivo conditions. The article aims to summarize the latest findings on a role of TRPM2 in endothelial cells. We also show some future perspectives for the application of TRPM2 inhibitors in cardiovascular system diseases.

## 1. Introduction

The transient receptor potential (TRP) melastatin-like subfamily member 2 (TRPM2; previously reported as TRPC7 or LTRPC2) is a non-selective calcium-permeable cation channel encoded by a *TRPM2* gene located on the human chromosome 21 [1]. Its expression has been confirmed in many mammalian tissues, such as bone marrow, spleen, lungs, heart, liver, macrophages, neutrophils, and endothelial cells [2]. There are several isoforms of TRPM2, including the short splice variant TRPM2-S, which is a shorter form of protein devoid of 4 of 6 carboxyl-terminal transmembrane domains. The interaction between full-length TRPM2 and TRPM2-S regulates Ca^2+^ influx in response to hydrogen peroxide, as confirmed in transfected 293T cells (human embryonic kidney) [3]. Because of the lacking C-terminal domain, TRPM2-S may function as an inhibitor of the full-length TRPM2 channel. Moreover, other splice variants were found in hematopoietic cells. It includes TRPM2-ΔN without amino acids 538–557 in the N-terminus, TRPM2-ΔC lacking amino acids 1292–1325 in C-terminus [4].

The functional TRPM2 channel is a tetramer. Its structure includes a nucleoside diphosphate binding fold located in the cytoplasmic carboxyl domain. It is responsible for the binding of nucleotide agonists and factors like adenosine 5′-diphosphoribose (ADPR), cyclic ADPR (cADPR), adenosine monophosphate (AMP), or nicotinamide adenine dinucleotide (β-NAD^+^) [4,5]. It is also stimulated in the conditions of enhanced oxidative stress. In turn, the N-terminal domain contains a calmodulin (CaM)-binding site, which is responsible for the regulation of TRPM2 by the calcium ions [6].

TRPM2 channel activity was so far connected with such events as a cellular response to ischemia–reperfusion injury, regulation of endothelial permeability, inflammation, development of cancer and degenerative diseases, or induction of cell death, including apoptosis and autophagy [2,7,8,9,10,11]. In the context of the endothelial layer, the impact of TRPM2 activation on barrier function, apoptosis, cell migration, angiogenesis, and transendothelial migration of the leukocytes seems to be particularly interesting. The article aims to summarize the latest findings on the role of TRPM2 in endothelial cells. We also show some future perspectives for the application of TRPM2 inhibitors in cardiovascular system diseases.

## 2. TRPM2 Structure and Activation Mechanisms

A functional form of the TRPM2 channel is a tetramer. Each monomer consists of an N-terminal region (approximately 800 amino acids in length), six transmembrane domains (S_1_–S_6_) with three extracellular loops, and a C-terminal domain. Both terminal domains are located inside the cell. The pore-forming loop is between S_5_ and S_6_. A characteristic feature of the N-terminus is the content of four TRPM subfamily homologous domains and an IQ-like motif responsible for the binding of CaM. The C-terminus contains a nudix-box sequence motif (NUDT9-H) [12]. The structure of the TRPM2 channel is shown schematically in Figure 1.

The best-known mechanism of regulation of the TRPM2 channel is the binding of ADPR to the NUDT9-H in the C-terminus [12]. The domain is also characterized by ADP hydrolase activity, which however is weak. Moreover, it seems that the prolonged interaction of ADPR with TRPM2 stimulates the channel. The intensification of enzymatic activity and the acceleration of the ADPR decomposition prevents channel opening [13]. The production of ADPR in the cells correlates with oxidative stress. Free radicals induce DNA damages, which stimulates poly-ADPR polymerase (PARP) to produce ADPR through the breakdown of NAD to poly(ADPR) and its subsequent degradation by the poly(ADPR) glycohydrolase (PARG) [14]. An alternative mechanism may be direct hydrolysis of NAD^+^ into nicotinamide and ADPR. However, the PARP-involving mechanism seems to be much more important [2]. In the interaction of NUTDH9-H and ADPR, the ADPR terminal ribose is crucial as ADP alone does not induce channel opening [15]. An interesting issue is also the co-activation of the channel by ADPR and calcium ions. The Ca^2+^-binding site is located close to the intracellular border of the TRPM2. Only the binding of calcium ions and ADPR together is sufficient to activate the channel [16]. In general, the functioning of TRPM2 appears to be closely dependent on calcium ions. Some of the studies also suggest that the channel may be activated by the action of Ca^2+^ alone [17]. An even more effective channel agonist appears to be 2′-deoxy-ADPR [5]. In vitro, it is formed from nicotinamide mononucleotide (NMN) and 2′-deoxy-ATP in a reaction catalyzed by cytosolic nicotinamide mononucleotide adenylyltransferase 2 (NMNAT-2) and nicotinamide adenine dinucleotide (NAD)-glycohydrolase CD38. As 2′-deoxy-ADPR induces 10.4-fold higher whole-cell currents compared to ADPR, it makes it a superagonist of the TRPM2 channel [5]. While in the case of ADPR and 2′-deoxy-ADPR, the situation seems quite clear, activation of TRPM2 by cADPR raises more controversy [18]. Kolisek et al. confirmed the channel opening only after administration of very high doses of cADPR. Interestingly, the simultaneous treatment with cADPR and ADPR showed a stronger effect than either of these substances alone, which may suggest their synergistic action [19]. It is also possible that cADPR is not a TRPM2 agonist at all and the effect observed in some of the studies is related to the easy degradation of cADPR to ADPR. Heiner et al. performed HPLC analysis of freshly prepared cADPR solutions and found that they contained approximately 25% ADPR. In turn, after purification, the addition of cADPR alone no longer induced intracellular ion influx in neutrophils [20]. On the contrary, Yu et al. observed effective TRPM2 activation by purified cADPR by the binding to the NUDT9-H [21]. Due to the inconsistency in the results, more reports on cADPR-dependent TRPM2 activation are necessary. It is also required to use purified cADPR and confirm the purity using, e.g., HPLC. A similar controversy is raised by the role of NAD as a TRPM2 agonist. Some reports indicate that NAD can directly activate TRPM2, while others suggest that it is due to its breakdown or contamination with ADPR [22,23,24]. Mechanisms of ligand-dependent TRPM2 activation were schematically presented in Figure 1.

Additionally, the opening of the TRPM2 channel pore is dependent on reactive oxygen species (ROS). Two alternative models of TRPM2 activation by ROS have been presented by Hara et al. and Wehage et al. [1,4]. The former points to the stimulation of TRPM2 in response to hydrogen peroxide by the action of β-NAD^+^ metabolites, while the latter suggests activation independent of ADPR. Wehage et al. used cells with a deletion in the C terminus (amino acids 1292–1325) and treated them with hydrogen peroxide or ADPR. They observed the successful stimulation of TRPM2 only in the H_2_O_2_-treated cells [4]. As the cells did not contain the functional NUDT9-H motif, they could not respond to the ADPR stimulation. It is possible that both mechanisms occur in parallel or not, depending on the splice variant of the channel.

Another described TRPM2 activation pathway is connected with the sensation of non-noxious warmth. Heat-dependent TRPM2 opening is an important mechanism in thermally-sensitive neurons as the channel passes the information regarding the temperature and stimulates cool-seeking behavior [25]. 

## 3. TRPM2 in Endothelial Permeability

The endothelium acts as a selective barrier that allows the transport of chosen substances between the blood and surrounding tissues. This function reflects in the structure of the endothelium, which is built of closely packed cells. The tightness of the endothelial layer is ensured by a network of intercellular connections consisting of the adherens and tight junctions. One of the symptoms of endothelial dysfunction is the appearance of spaces between the cells resulting from the loosening of the structure of intercellular connections [26]. The development of endothelial dysfunction is strictly connected with inflammatory reactions. One of the mediators generated at the site of inflammation are ROS, which induce the development of endothelial dysfunction. Even if the exact course of the ROS action in endothelial cells remains elusive, it is known that they activate membrane ion channels, resulting in an intracellular influx of calcium ions. The effect is gap formation and enhanced endothelial permeability [27,28].

As showed by Hecquet et al., one of the pivotal factors for ROS-induced permeability in endothelial cells is the TRPM2 channel [2,29]. It participates in the regulation of human pulmonary artery endothelial (HPAE) cell permeability in response to H_2_O_2_. Treatment of HPAE cells with non-lytic doses of hydrogen peroxide resulted in a rapid intracellular influx of calcium ions. It was associated with a loosening of the intercellular junction network manifested as a reduction in transendothelial electrical resistance (TER). This effect was counteracted by both transfection with TRPM2 siRNA and preincubation with a TRPM2-blocking antibody. The effect of H_2_O_2_ was also limited by the use of PARP inhibitors and the overexpression of TRPM2-S [2]. In turn, TRPM2-L overexpression additionally facilitated intracellular gap formation. Although different methods of inhibiting TRPM2 activation resulted in similar effectiveness in limiting the calcium current, the reduction in TER was inhibited by just 50%. It suggests the existence of a Ca^2+^-independent mechanism of H_2_O_2_ action [2].

Interestingly, the authors also confirmed the involvement of protein kinase C alpha (PKCα) in the activation of TRPM2 [2]. Application of PKCα inhibitor (Gö6976) or PKCα siRNA led to partial inhibition of the intracellular influx of Ca^2+^. It is also possible that PCKa is one of the effectors, as its activation leads to VE-cadherin disassembly in response to factors such as thrombin [30]. It is, therefore, possible that the increase in endothelial permeability in the course of oxidative stress is related to a similar mechanism.

An issue closely related to endothelial permeability is leukocyte transmigration. Due to the local reduction in the barrier function of the endothelium, leukocytes, in response to the inflammatory mediators, can quickly pass from the blood to the inflamed tissues between the endothelial cells. Activated neutrophils are characterized by increased production of ROS, which has an antibacterial effect, but also induces the changes observed in endothelial dysfunction. As showed by Mittal et al., TRPM2 channel activation is essential for neutrophil transmigration [31]. In the studies comparing the response to lipopolysaccharide (LPS) between mice with conditionally deleted TRPM2 expression in endothelial cells (*Trpm2^iΔEC^*) and wild-type mice, the authors observed significantly reduced polymorphonuclear neutrophils (PMNs) transmigration and mortality in the *Trpm2^iΔEC^* group. The crucial element for TRPM2 activation was ROS generated by PMNs, which then stimulated PARP necessary for the formation of ADPR. The results were confirmed in human lung microvascular endothelial cells (HLMVECs) with decreased expression of TRPM2 or PARP. Silencing of both targets attenuated the response of cells to the H_2_O_2_ and PMNs-dependent activation. Additionally, stimulation of mouse lung microvascular endothelial cells with PMNs pretreated with diphenyliodonium (NADPH oxidase inhibitor) also resulted in limited calcium influx. The key phenomenon for the transmigration of leukocytes is the loosening of the structure of intercellular connections between endothelial cells, which allows immune system cells to squeeze between these thin cells that build the first layer of blood vessels. In anti-inflammatory conditions, the barrier function of the endothelium is ensured by the proteins involved in the formation of adherens and tight junctions. Thus, a necessary element of transmigration is the degradation of cell junctions. In endothelial cells, transfection with TRPM2 siRNA suppressed the phosphorylation of vascular endothelial (VE)-cadherin at Y731 in response to the stimulation with PMNs. Phosphorylation of the VE-cadherin is necessary for its disassembly and internalization [31].

In turn, a protein crucial for the assembly of tight junctions is, for example, zonula occludens-1 (ZO-1). As shown by Wang et al., TRPM2-dependent calcium influx affects the stability of ZO-1 in human lung microvascular endothelial cells exposed to particulate matter (PM) [32]. Treatment of endothelial cells with PM led to the reduction in TER, which suggests increased permeability. However, another PM-induced effect was a decrease in the level of ZO-1 and ZO-2. Interestingly levels of VE-cadherin and β-catenin were not affected. The observed phenomenon was connected with ROS production as both ROS scavenger N-acetylcysteine (NAC) and PEG-catalase, which degrades hydrogen peroxide, prevented ZO-1 disassembly. It was proven that in the case of human lung microvascular endothelial cells, exposure to PM leads to the ROS-dependent TRPM2 activation, which results in rapid calcium current as the application of both TRPM2 siRNA and anti-TRPM2 antibody resulted in the inhibition of ZO-1 disassembly. In turn, increased calcium concentration activates calpain, a calcium-dependent protease responsible for ZO-1 degradation. Pretreatment of cells with calpain inhibitors and calpeptin prevented endothelial barrier disruption [32].

## 4. TRPM2 in Endothelial Cell Death

Numerous reports indicate that TRPM2 is involved in the ROS-induced death of hematopoietic cells, neurons, and vascular endothelium [10,33,34,35]. Ca^2+^ entry, facilitated among others by TRPM2, leads to its rapid and excessive accumulation. In turn, disturbances in intracellular calcium homeostasis contribute to cell death. Though apoptosis is important for the maintenance of homeostasis, injury repair, and organ development, in endothelium it also facilitates pathological changes connected with inflammatory reaction and vascular diseases.

In heart microvessel endothelial cell line H5V, one of the observed effects of H_2_O_2_ treatment was the decrease in metabolic activity of cells observed in the MTT assay. Caspase levels examination by electrophoretic DNA fragmentation showed that the H_2_O_2_ caused a decrease in cell viability, which was related to apoptosis. However, the authors did not analyze the percentage of necrotic cells, which may be identified by a characteristic smear visible after the electrophoretic separation of DNA [10]. On the other hand, the use of TRPM2-specific shRNA resulted in the reduction of caspase 3, 8, and 9 activation, which indicates an inhibition of apoptosis. Additionally, it is also possible that TRPM2 is involved in the induction of cell death by one of the proinflammatory cytokines tumor necrosis factor alpha (TNFα). 36 h treatment of H5V cells with 10 ng/mL of TNFα resulted in a decrease in cell viability. This effect was limited after using TRPM2 blocking antibody (TM2E3) or TRPM2-specific shRNA [10].

Similar results were obtained by Hecquet et al. in the studies on human pulmonary artery endothelial cells (HPAEC). Treatment of HPAEC with hydrogen peroxide resulted in a significant increase in the percentage of apoptotic cells. However, the effect was abolished after using TRPM2 siRNA or an anti-TRPM2 blocking antibody. The results were confirmed in in vivo experiments, in the case of which more endothelial cells undergoing apoptosis were observed for wild type than *TRPM2^−/−^* mice. Most interestingly, the author focused also on the interaction between shorter splice variant TRPM2-S (dominant negative form) and full-length TRPM2-L. They observed that PKCα activation followed by the phosphorylation of the TRPM2-S at Ser39 caused its disassociation from the TRPM2-L, which released the inhibitory effect and led to the rapid calcium influx [2]. Moreover, mutation at Ser39 of the TRPM2-S blocked its phosphorylation, which resulted in limited Ca^2+^ current and prevented cell apoptosis. It is consistent with results obtained for Jurkat T-lymphocytes and K562 myeloid leukemia cell line, which express TRPM2-S at a very low level and are unresponsive for TRPM2 activation [35,36].

Furthermore, TRPM2 may be involved in high glucose-induced mitochondrial fission in endothelial cells [37]. The process of mitochondrial fission is strictly connected with apoptosis [38]. One of the consequences of diabetes is a high concentration of circulating glucose. It leads to the production of ROS, which affects endothelial cells and whole circulatory system. Abuarab et al. showed that inhibition of TRPM2 using its inhibitor or TRPM2 silencing RNA blocked the mitochondrial fission in HUVEC cells [37]. Similarly, primary endothelial cells obtained from TRPM2 knockout mice were less susceptible to high glucose-induced mitochondrial fission compared to wild-type mice. Moreover, the observed effect was not connected with store-operated Ca^2+^ entry as the silencing of stromal interaction molecule 1 and Orai-1 did not affect the high glucose-induced calcium influx. The authors proved that the observed effect was connected with ROS production stimulated by high glucose concentration as preincubation with *N*-acetyl-cysteine (ROS scavenger) prevented mitochondrial fragmentation [37].

It is also worth mentioning that TRPM2 may contribute to the type II programmed cell death called autophagy. Although no reports are showing TRPM2 involvement in autophagy induction in endothelial cells, there is evidence of its participation in this type of cell death in vascular smooth muscles and brain pericyte injury [11,39]. In primary cultured mouse aortic smooth muscle cells (mASMCs) TRPM2 expression was found in lysosomes/autolysosomes resulting from nutrients starvation. Both autophagic response and lysosomal/autolysosomal acidification were reduced after TRPM2 knockout [11]. In turn, in the human brain vascular pericytes, zinc oxide nanoparticles (ZnO-NP)-induced autophagy was accompanied by increased TRPM2-S expression. Knockout of TRPM2 preserved the structure of brain microvessels in mice and prevented vascular injury induced by ZnO-NP [39]. Thus, the TRPM2-dependent influx of calcium ions appears to have a broad effect on the circulatory system. However, in the case of autophagy, the matter remains controversial as it can be cytoprotective or act as a cell death mechanism [40].

## 5. TRPM2 in Endothelial Cell Migration and Angiogenesis

Neovascularization under ischemic conditions requires endothelial cells, ROS, and growth factors. Vascular endothelial growth factor (VEGF) in endothelial cells propels their migration, proliferation, and disassembly of adherens junctions, as well as ROS production. All of these elements are necessary for the angiogenesis process [41]. Another very important factor for the occurrence of angiogenesis are Ca^2+^ [42,43]. Mittal et al. showed that VEGF stimulates ROS-dependent activation of TRPM2 [44]. In the cells treated with PARP inhibitor (3-aminobenzamide), VEGF-induced Ca^2+^ influx was limited comparing to the untreated human pulmonary artery endothelial cells. The same result was observed for *TRPM2*−/− mice compared to wild-type mice. They also showed that TRPM2, after binding agonist, recruits c-Src and activates it. It results in VE-cadherin phosphorylation, internalization, and enhanced endothelial cell migration in vitro, as observed in wound healing assay. Moreover, in vivo experiments confirmed the pivotal role of TRPM2 in angiogenesis and post-ischemic neovascularization. In both in vivo Matrigel plug assay and ex vivo aortic ring assay, the authors observed impaired angiogenesis only in the case *of TRPM2*−/− mice [44]. These findings seem particularly interesting considering TRPM2 involvement in the ischemic/reperfusion injuries in other cells like neurons or hepatocytes [45,46]. Effects induced by TRPM2-dependent calcium ions influx are summarized in Figure 2.

## 6. TRPM2 Inhibitors in Clinical Application and Further Perspectives

Reports included in this review highlight the pivotal role of TRPM2-dependent calcium influx in the permeability, cell death, and migration of endothelial cells in both in vitro and in vivo conditions. However, currently, there are some problems with TRPM2 inhibitors that have to be resolved before their clinical application. Although numerous TRPM2 inhibitors are known, they are unable to target the TRPM2 channel with appropriate specificity. The examples are N-(p-Amylcinnamoyl) anthranilic acid, widely used in research, which additionally affects TRPM8, TRPC6, TPPV1, TRPC3, or flufenamic acid inhibiting TRPC3, TRPC7, TRPM2–5, TRPM7/8, TRPV1, TRPV3/4 [47]. However, some efforts are made to enhance the specificity of ACA by using its derivatives [7]. Moreover, some of the substances are characterized by low inhibitory effects or poor membrane permeabilization properties [48]. In turn, some of the promising inhibitors lacking stability in in vivo conditions [49]. Recently, the influence of various substances of natural origin, such as resveratrol or curcumin in the context of TRPM2 inhibition, has been intensively studied [50,51]. However, so far, no similar studies have been performed on endothelial cells. The substances of natural origin will likely encounter the same problems as other inhibitors, i.e., lack of specificity and stability [52]. Some of the potential inhibitors of TRPM2 are included in Figure 1.

Since the opening of the TRPM2 channel is observed in the course of increased oxidative stress, the logical approach seems to be the use of antioxidants. However, in this case, the results remain inconclusive. As showed by Nazıroğlu and Lückhoff, treatment of Chinese hamster ovary (CHO) cells with antioxidants such as ascorbic acid (vitamin C), alpha-tocopherol (vitamin E), or glutathione did not prevent or even weaken the H_2_O_2_-induced TRPM2 opening and following Ca^2+^ current [53]. However, this study has several limitations, some of which have also been described by the authors. The first problem is the optimization stage of the study. The doses of the antioxidants selected by the researchers could be inadequate. The concentration of hydrogen peroxide used in the study is also significant here. Due to the inability to activate TRPM2 with low concentrations of H_2_O_2_, mM concentrations were used in the study. For most cells, severe changes are induced already with nM or µM oxidant concentrations. In this case, the doses of antioxidants should also be increased accordingly. Moreover, later in the study, cells transfected with the expression plasmid were used to increase TRPM2 expression. The obtained results are therefore of little importance in the clinical context. It might be more valuable to use cells that naturally are characterized by a relatively high TRPM2 expression in similar research. An important variable here is also the experiment design and treatment regimen. Whether and how long cells should be pretreated with the selected compounds, or maybe a better solution would be the simultaneous administration of an antioxidant and a substance that intensifies oxidative stress. The authors also point out that a mixture of different antioxidants could be more effective, as it better reflects the in vivo conditions [53]. In turn, other studies indicate an important role of antioxidants in the functioning of the TRPM2 channel. In the case of human lung microvascular endothelial cell treatment with NAC, which also belongs to the group of antioxidants, or EUK-134, a ROS scavenger, attenuated the negative effects accompanying endothelial layer disruption connected with the activation of TRPM2 ion channel [32]. 

An interesting approach may also be the upregulation of the TRPM2-S dominant negative shorter splice variant of TRPM2 lacking the NUDT9-H domain. TRPM2-S interacts with the full-length isoform and prevents the assembly of the functional homotetrameric channels. As endothelial cells naturally express TRPM2-S, it may turn out to be the safest method. However, currently, no protocols allow effective TRPM2-S upregulation and can be used in the clinical context.

Another option is inhibition of PARP, which prevents the formation of TRPM2 which prevents the formation of ADPR as an activator of TRPM2 channels. Although some PARP inhibitors are already approved by Food and Drug Administration (FDA) and European Medicine Agency (EMA), they are currently used mostly in cancer treatment. However, there is strong evidence suggesting that PARP inhibitors improve endothelial function impaired by the aging process [54,55,56]. However, due to the involvement of PARP in such important and complex processes as DNA damage repair, the observed effect of the inhibitors will not be at all or will be only partially related to the blockade of ADPR-induced TRPM2 activation. Chosen PARP inhibitors were included in Figure 1.

## 7. Conclusions

The TRPM2-dependent influx of calcium ions plays a pivotal role in endothelial cell function. It regulates permeability, transendothelial migration of the leukocytes, cell death, migration, and angiogenesis. Therefore, it may be an attractive therapeutic target in the context of endothelial dysfunction induced by increased oxidative stress. However, there are still some issues to be clarified. It is unclear whether compounds such as cADPR or NAD are ligands of the channel and lead to its opening or not. If so, the exact binding mechanism is also unknown. This knowledge could lead to the design of new and better channel inhibitors. However, to enable the application of TRPM2 inhibitors in clinical practice, it is necessary to increase their specificity, membrane permeability, and stability. Another unclear issue is whether antioxidants can prevent TRPM2-dependent changes in the endothelial barrier. Thus, research on endothelial cells treated with antioxidants may also be beneficial in the clinical context.

## Figures and Tables

**Figure 1 ijms-22-07635-f001:**
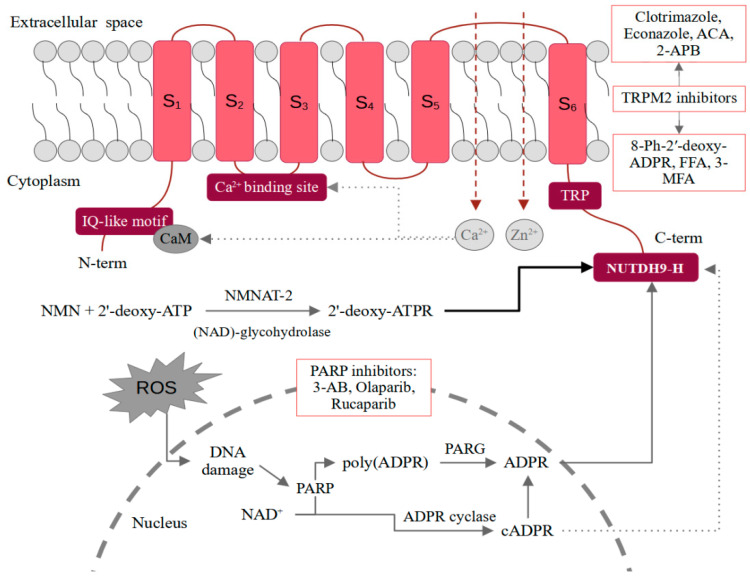
Mechanisms of ligand-dependent TRPM2 activation together with potential inhibitors. 2-APB—2-aminoethoxydiphenyl borate; 3-AB—3-Aminobenzamide; ACA—*N*-(*p*-amylcinnamoyl) anthranilic acid; ATP—adenosine triphosphate; ATPR—ATP-ribose; cATPR—cyclic ATP-ribose; C-term—C-terminal domain; CaM—calmodulin; FFA—flufenamic acid; MFA—mefenamic acid; N-term—N-terminal domain; NAD—nicotinamide adenine dinucleotide; NMN—nicotinamide mononucleotide; NMNAT-2—nicotinamide mononucleotide adenylyltransferase 2; PARG—poly(ADPR) glycohydrolase; PARP—poly-ADPR polymerase; ROS—reactive oxygen species; TRP —transient receptor potential; TRPM2—TRP melastatin-like subfamily member 2.

**Figure 2 ijms-22-07635-f002:**
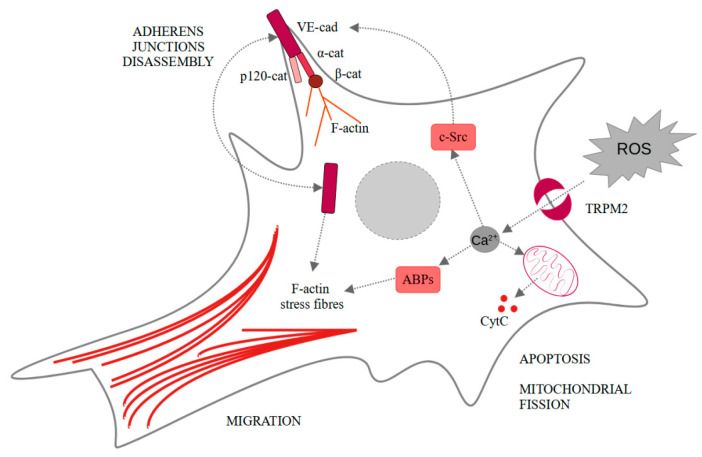
Intracellular changes induced in endothelial cells by the TRPM2-dependent calcium influx. α-cat—α-catenin; β-cat—β-catenin; ABPs—actin-binding proteins; CytC—cytochrome C; F-actin—fibrillar actin; p120-cat—p120-catenin; ROS—reactive oxygen species; TRPM2—transient receptor potential melastatin-like subfamily member 2.

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
