# Peer review of "The Role of TRPM2 in Endothelial Function and Dysfunction"

_ijms, 2021, doi:10.3390/ijms22147635_

Round 1
Reviewer 1 Report
In the manuscript by Zielinska et al. the authors summarize recent data on TRPM2 function in endothelial cells. The review is comprehensive and cites the recent literature. However, English phrases and the figures need to be improved:
- The whole text needs to be corrected by a native speaker. A few examples:
Line 56: “summarize the latest findings regarding TRPM2 role in endothelial cells.”
Change to “summarize the latest findings on a role of TRPM2 in endothelial cells”
Line 57: “for the application of TRPM2 inhibitors in conditions connected with the cardiovascular system”. Change to “for the application of TRPM2 inhibitors in the cardiovascular system”
Line 70: “Although the domain is also characterized by ADP hydrolase activity, it is relatively weak”. Change to “The domain is also characterized by ADP hydrolase activity, which however is weak. “
Line 109: “Additionally, the TRPM2 opening is dependent on reactive oxygen species (ROS) availability.” Change to “Additionally, the opening of the TRPM2 channel pore is dependent on reactive oxygen species (ROS).”
Line 111: “point” change to “points”
Line 152: “However” change to “Along this line”
Line 177: delete “in the case of of”
Line 218: “authors did not analyze the percentage of necrotic cells , the presence of which may be visible suggested by”. Change to “authors did not analyze the percentage of necrotic cells, which may be identified by”
Line 231: “What’s interesting the authors focused”. Change to “Most interestingly, the author focused”
Linie 252: N-acetyl cysteine is rather a ROS “scavenger” than a ROS “inhibitor”.
Line 337: “prevents the formation of TRPM2 main ligand ADPR.” Change to “ which prevents the formation of ADPR as activator of TRPM2 channels”.
- Where is the Ca2+ binding site exactly located? It is mentioned in the text (line 83) that “a Ca2+ binding site is located close to the intracellular border of the TRPM2” (channel). Please indicate the (proposed) Ca2+ and calmodulin (IQ-like motif) binding sites in Figure 1.
- Figure 1: Please insert how DNA damage is activating TRPM2.
- Figure 2: Define target molecules, which are activated by Ca2+ ions through TRPM2 channels (arrows with (stippled) lines from Ca2+ to the targeted molecules).
Author Response
Dear Reviewer,
Thank you for your valuable comments. All of them were carefully considered by our research team and the following changes were introduced:
- All of the corrections regarding the language were included. Additionally, the whole manuscript was improved by a native English speaker.
- Figures 1 and 2 were modified following the suggestions.
Reviewer 2 Report
The review manuscript by Zielińska et al. aimed to update the knowledge about the role of TRPM2 in endothelial biology. This manuscript is of interest; however, the parts regarding the role of TRPM2 in the regulation of autophagy and mitochondria could be addressed more for increasing the scientific richness of this manuscript. In addition, it will be nice to add new figures for above mentioned review points.
Author Response
Dear Reviewer,
Thank you for your valuable comments. They were carefully considered by our research team and the following changes were introduced:
- The involvement of TRPM2 in the autophagy of cells connected with the functioning of the circulatory system was described in more detail.
- Mitochondrial fission was mentioned as the effect of TRPM2-dependent calcium influx in Figure 2.
- Unfortunately, we were unable to expand the fragment regarding TRPM2, mitochondrial fission, and endothelial cells as there is only a single report confirming the connection between them. There are also no reports on TRPM2 involvement in endothelial cells autophagic response. However, TRPM2 activation is connected with the autophagy of other components of blood vessels like vascular smooth muscle cells or brain vascular pericytes. Both of these reports were included in the main text. Due to the limited amount of data regarding the relationship between TRPM2, endothelial cells, and autophagy or mitochondrial fission, we are unfortunately not able to depict their course in the form of figures.
Round 2
Reviewer 1 Report
Thank you for the revised version. All points of my review were sufficiently adddressed, but please remove the old version of Figure 2.
Reviewer 2 Report
Authors have addressed my comments.